# Impedance Spectroscopy Measurements of Ionomer Film Oxygen Transport Resistivity in Operating Low-Pt PEM Fuel Cell

**DOI:** 10.3390/membranes11120985

**Published:** 2021-12-16

**Authors:** Tatyana V. Reshetenko, Andrei Kulikovsky

**Affiliations:** 1Hawaii Natural Energy Institute, University of Hawaii, Honolulu, HI 96822, USA; 2Theory and Computation of Energy Materials (IEK-13), Institute of Energy and Climate Research Forschungszentrum Jülich GmbH, D-52425 Jülich, Germany; A.Kulikovsky@fz-juelich.de

**Keywords:** PEMFC impedance, modeling, cathode catalyst layer, ionomer film transport resistivity

## Abstract

The work presents a model for local impedance of low-Pt proton exchange membrane fuel cells (PEMFCs), including cathode pore size distribution and O_2_ transport along pores and through a thin ionomer film covering Pt/C agglomerates. The model was applied to fit the local impedance spectra of low-Pt fuel cells operated at current densities from 100 to 800 mA cm^−2^ and recorded by a segmented cell system. Assuming an ionomer film thickness of 10 nm, the fitting returned the product of the dimensionless Henry’s constant of oxygen dissolution in ionomer *K_H_* by the oxygen diffusivity *D_N_* in the ionomer (*K_H_D_N_*). This parameter allowed us to determine the fundamental O_2_ transport resistivity $R_{N}$ through the ionomer film in the working electrode under conditions relevant to the realistic operation of PEMFCs. The results show that variation of the operating current density does not affect $R_{N}$, which remains nearly constant at ≃0.4 s cm^−1^.

## 1. Introduction

At the end of 1838, Christian Friedrich Schoenbein immersed two platinum wires surrounded by glass vessels in liquid electrolyte and connected the electrodes to a Volta battery. Experiments with this simple system led him to the discovery of the fuel cell effect [1]. Ever since, due to its unique catalytic properties, Pt has been the material of choice for low temperature proton exchange membrane fuel cells (PEMFCs).

Unlike Schoenbein’s Pt wires, the modern PEMFC cathode is a porous composite material containing Pt nanoparticles on a carbon support, and a decrease in Pt loading in this system is a high-priority task for fuel cell technology. A decade ago it appeared that Pt reduction was a more complicated problem than anticipated, leading to a drop in fuel cell performance, which exceeded the expected loss due to a lowering of the Pt active area [2,3].

Further studies have revealed that the observed performance decline is caused by increased mass transport losses [3,4,5,6,7,8,9,10]. In the cathode catalyst layer (CCL), the oxygen pathway to the oxygen reduction reaction (ORR) site includes diffusive transport along the void pore, dissolution in the thin ionomer film and diffusion through the film to the Pt surface (Figure 1). Qualitatively, a lower number of ORR sites in the low-Pt electrode means higher oxygen and proton flux per site, making the ionomer film quite a significant barrier for oxygen transport to the site. Thus, in high-Pt cells, the effects of oxygen transport through the film are marginal, while in low-Pt cells, this transport gives quite a substantial contribution to the potential loss. Another explanation of the over-linear transport loss by the vulnerability of low-Pt electrodes to flooding has been suggested in [11].

The limiting current method is an established technique for the determination and analysis of O_2_ mass transport resistance in PEMFC electrodes [3,4,9,12,13,14,15,16]. A variation of oxygen concentration, total gas pressure or oxygen balance gas results in the separation of: (1) transport resistance due to molecular diffusion in gas channels and large pores; and (2) resistance originating from Knudsen diffusion in fine pores and diffusion from ionomer and liquid water films [3,4,9,14,15]. The effective thickness of the ionomer films estimated using O_2_ mass transport resistance was found to be significantly higher than that observed by transmission electron microscopy of 1–10 nm [6,7,9]. This finding suggested that the physical properties of the thin ionomer films in confinements such as water uptake and elastic modulus differed from the properties of bulk ionomer, and the additional interaction of ionomer and Pt negatively affected mass transport [6,7,9,17,18,19,20,21,22]. In spite of the fact that the limiting current approach is widely used for PEMFCs characterization, its implementation typically requires a cell with an active area of 1–5 cm^2^ and high stoichiometry of reagents to reach currents relevant for PEMFCs. The chosen conditions seem to be not quite representative for realistic fuel cell operation. The application of a segmented cell system and the distribution of limiting currents along a flow field can resolve this issue, since a cell with a high active area (>50 cm^2^) can be used and operated at realistic conditions [14,23,24]. However, it should be mentioned that a segmented cell system is not a widely available equipment, which makes its application limited. Therefore, developing methods and models for the determination of O_2_ mass transport parameters at conditions which are relevant for realistic PEMFCs operation and easily available for researchers is a very important task for the fuel cell community.

Currently, electrochemical impedance spectroscopy (EIS) seems to be the only easily accessible tool which allows us to obtain an *in-situ* diagnostic of operating PEMFCs. Moreover, the EIS method is very sensitive to oxygen transport processes in a fuel cell. Recently, we have reported a model for cell impedance which takes into account oxygen transport through the ionomer film [25]. The model links, in a 1d + 1d manner, the oxygen diffusion equations along the cylindrical void pore and through the ionomer film surrounding the pore. In this work, the model is extended for the processing of local impedance spectra, considering pore size distribution (PSD) in the electrode. To the best of our knowledge, this is the first impedance model of an operating low-Pt fuel cell that considers the PSD of the CCL.

This work is a continuation of our previous studies [25]. We determined the ionomer film oxygen transport resistivity $R_{N}$ based on a fitting of localized impedance spectra acquired from 10 segments of a low-Pt cell in a segmented cell setup. The experimental PSD of the cathode was approximated by the pores of nine characteristic radii. Assuming a film thickness of 10 nm, from each experimental run we obtained 10 local values of the product *K_H_D_N_*, where *K_H_* and *D_N_* are the Henry’s constant for oxygen solubility and oxygen diffusivity in the ionomer film, respectively. The mean over the cell surface value of *K_H_D_N_* and the standard deviation allowed us to calculate a statistically significant $R_{N}$ for every cell current density fixed in the experiments. In the range of 100 to 800 mA cm^−2^, it was determined that $R_{N}$ ≃ 0.4 s cm^−1^ is nearly independent of cell current density.

## 2. Materials and Methods

### 2.1. Electrochemical Evaluation

The impedance spectra were acquired using a segmented cell system designed at the Hawaii Natural Energy Institute (HNEI). The system includes a custom-built test station, segmented cell hardware, current/voltage sensors and a PXI data acquisition system [26]. The test station controls gas flow rates, humidification and back pressure, cell and gas line temperature and the overall cell current or voltage. The segments’ currents are measured but not controlled in order to mimic the experimental conditions of a non-segmented cell.

In this work, we used the Gore PRIMEA (A510.1/M715.18/C510.1) catalyst coated membrane (CCM) with an active area of 100 cm^2^. The Pt/C catalyst loading for anode and cathode was 0.1 mg_Pt_ cm^−2^. The electrode thickness varied from 3 to 4 μm. Sigracet 25BC, with a thickness of 220–230 μm, was employed as the gas diffusion layer (GDL). To keep an optimal membrane electrode assembly (MEA) compression ratio of 25–30%, we applied Teflon gaskets with a thickness of 125 μm on both sides of the cell.

Electrochemical measurements were performed using modified 100 cm^2^ hardware which had segmented and non-segmented flow fields. The segmented flow field and current collector were separated into 10 segments with an area of 7.6 cm^2^, connected in order from segment 1, located at the gas inlet, to segment 10, at the outlet. Due to the segmentation of the cathode flow field, the resulting active MEA area was 76 cm^2^. The segmented and unchanged flow fields had the same 10-channel parallel serpentine design and were arranged in co-flow configuration. Details of the segmented cell system operation are given in [26,27].

Impedance spectra were measured under galvanostatic control of the whole cell with 11 steps/decade; the cell operating conditions are listed in Table 1. The frequency range was from 10 kHz to 0.05 Hz. The amplitude of current perturbation was adjusted to obtain a cell voltage response of 10 mV or less. The sampling rate of the current/voltage response from 10 segments was 1 MHz.

### 2.2. Impedance Model

The cell used in the experiments has been modeled as a cell with the straight cathode channel separated into 10 segments (Figure 2a). The model of the local segment impedance *Z_seg_* is based on a transient CCL performance model, which includes a mass conservation equation for oxygen transport in the GDL. The CCL is modeled as a system of cylindrical pores; each pore is inserted into a coaxial Pt/C layer and separated from this layer by a thin ionomer film (Figure 2b). Oxygen is transported along the pore axis and in the radial direction through the ionomer film to the ionomer/Pt interface, where the oxygen reduction reaction occurs. Model equations have been described in details in [25]; here, for completeness, we list the basic equations.

### 2.3. Model Equations

Oxygen transport along the void pore (Figure 2b) is described by the transient diffusion equation:(1)$$\frac{\partial c}{\partial t}-D_{p}\frac{\partial^{2}c}{\partial t^{2}}=\frac{2N_{N,p}}{R_{p}},\;{\left.\frac{\partial c}{\partial x}\right|}_{x=0}=0,\;c\left(l_{t}\right)=c_{1}$$
where *c*_1_ is the oxygen concentration at the CCL/GDL interface, *l_t_* is the CCL thickness (the pore length), *D_p_* is oxygen diffusion coefficient in the pore,
(2)$$N_{N,p}=D_{N}{\left.\frac{\partial c_{N}}{\partial r}\right|}_{r=R_{p+}}$$
is the oxygen flux in the ionomer film at the pore/film interface, and *c_N_* is the oxygen concentration in ionomer film.

Oxygen transport through the ionomer film along the radial direction is described by:(3)$$\frac{\partial c_{N}}{\partial t}-\frac{D_{N}}{r}\frac{\partial }{\partial r}\left(r\frac{\partial c_{N}}{\partial r}\right)=0,\;c_{N}\left(R_{p}\right)=K_{H}c\left(x\right),\;D_{N}{\left.\frac{\partial c_{N}}{\partial r}\right|}_{r=R_{m}}=\;-\frac{i_{*}R_{p}}{2\left(4F\right)}\left(\frac{c_{N,m}}{c^{in}}\right)\exp\left(\frac{\eta }{b}\right)$$
where *K_H_* is the dimensionless Henry’s constant (mol/mol) for oxygen solubility in ionomer, *D_N_* is the oxygen diffusivity in the ionomer film, *R_m_* is the Pt/C tube radius (Figure 2b), $c_{N,m}\equiv c_{N}\left(R_{m}\right)$, $c^{in}$ is the reference (inlet) oxygen concentration, *b* is the ORR Tafel slope, and $i_{*}$ is the ORR exchange current density.

Proton current conservation in the ionomer film and the Ohm’s law for this current lead to an equation for the positive by convention ORR overpotential η:(4)$$C_{dl}\frac{\partial \eta }{\partial t}-\sigma_{p}\frac{\partial^{2}\eta }{\partial x^{2}}=-i_{*}\left(\frac{c_{N,m}}{c_{h}^{in}}\right)\exp\left(\frac{\eta }{b}\right)$$
where *C_dl_* is the double layer volumetric capacitance, and *σ_p_* is the ionomer film proton conductivity.

Oxygen transport in the gas diffusion layer is described by:(5)$$\frac{\partial c_{b}}{\partial t}-D_{b}\frac{\partial^{2}c_{b}}{\partial x^{2}}=0,\;{\left.\frac{\partial c_{b}}{\partial x}\right|}_{x=l_{t+}}={\left.\frac{\partial c}{\partial x}\right|}_{x=l_{t-}},\;c_{b}\left(l_{t}+l_{b}\right)=c_{h}$$
where *c_b_* is the oxygen concentration in the GDL, *D_b_* and *l_b_* are the GDL oxygen diffusivity and thickness, respectively, and *c_h_* is the oxygen concentration in the air channel. The latter value obeys to plug-flow mass transport equation along the channel coordinate *z*:(6)$$\frac{\partial c_{h}}{\partial t}+v\frac{\partial c_{h}}{\partial z}=-\frac{D_{b}}{h}{\left.\frac{\partial c_{b}}{\partial x}\right|}_{x=l_{b}}$$
where *v* is the air flow velocity and *h* is the channel depth.

As can be seen, transient diffusion equations for oxygen transport along the pore, Equation (1), and through the ionomer film, Equation (3), are linked in a 1d+1d manner. The ORR rate at the Pt/C interface is described by the Tafel law. The proton charge conservation equation along the ionomer film, Equation (4), takes into account finite proton conductivity of the film and charging/discharging of the double layer. Radial oxygen flux in the film at the pore/film interface represents a sink in the oxygen mass transport equation along the void pore [25].

The total oxygen flux along the void pore is proportional to the square of the pore radius, while this flux through the ionomer film covering the side surface of the pore is proportional to the first power of the radius. Evidently, the electrode O_2_ transport properties strongly depend on the PSD [28,29,30]. The PSD of a PRIMEA Gore CCM was presented in [31]. Since in the current work we applied the PRIMEA CCM as well, the reported PSD adequately described the pore size spectrum of the CCL under the study. To accelerate calculations, the PSD measured in [31] has been approximated by a stepwise function with the nine characteristic radii (Figure 3).

The pore radii and their relative fractions in the electrode volume calculated from the experimental PSD are depicted in Table 2. The CCL impedance *Z_ccl,n_* in the *n*-th segment has been calculated as an impedance of these parallel pores:(7)$$\frac{1}{Z_{ccl,n}}=\sum_{k=1}^{9}\frac{w_{k}}{Z_{k,n}}$$
where $Z_{k,n}$ and $w_{k}$ are the pore impedance and volume fraction, respectively, of the *k*-th pore in the spectrum.

Linearization and Fourier-transform of the transport equations lead to a system of equations for the AC perturbation amplitudes in the *ω*-space [25]. This system is solved numerically for every individual segment, taking into account transport of oxygen concentration perturbations with the channel flow (Figure 4). Air flow in the cathode channel transports local oxygen concentration perturbations $c_{h}^{1}$ to the segments located downstream. Thus, the local $c_{h}^{1}$ sums up with the transported perturbations, which affect the local impedance.

A custom parallel Python code was developed using the message passing interface library. Calculated segment impedance *Z_seg_* was fitted to the local experimental spectrum using the SciPy least-squares procedure *least_squares* with the *method = ‘trf’ option* [32]. The nine-pore model described above for each of the 10 segments requires 90 cores on a parallel cluster. Detailed description of the parent whole-cell impedance model can be found in [25].

## 3. Results and Discussion

The local impedance spectra were measured at the operating conditions listed in Table 1. Figure 5 demonstrates experimental and fitted Nyquist spectra for the current density of 400 and 800 mA cm^−2^. Typical impedance spectra for a PEM fuel cell are represented by a semicircle at high-frequency, reflecting a charge transfer, proton and oxygen transport within the cathode layer, and a low-frequency (LF) arc caused by oxygen transport in the channel [33,34]. The local EIS results show an increase in the LF arc size from the cell inlet (segment 1) to the outlet (segment 10) and clearly illustrate the effects of O_2_ depletion along the flow field (Figure 5a). At the outlet segments 8 to 10, the LF part of the spectra becomes noisy, and the quality of the fitting decays (Figure 5). However, this arc represents oxygen transport in the gas channel, while the characteristic frequency of ionomer film impedance is at least an order of magnitude higher [35]. Overall, the results show that the proposed model ensures a good fitting of the local EIS data at low and high current densities.

The model equations include pore radius *R_p_* and “metal” radius *R_m_*; the film thickness is $l_{N}=R_{m}-R_{p}$ (Figure 2b). Code testing shows that the fitting is more sensitive to the ratio *l_N_/D_N_*, rather than to *l_N_* and *D_N_* separately. On the other hand, equations in Section 2.3 show that *D_N_* appears in the model only as a product *K_H_D_N_*, where *K_H_* is the dimensionless Henry’s constant for oxygen solubility in the ionomer film. To reliably obtain the parameter *K_H_D_N_* from the fitting, the ionomer film thickness was fixed at *l_N_* = 10 nm. This thickness value agrees well with our previous measurements [25] and the other literature sources [6,7,9,36,37,38]. The starting value for iterations of *K_H_D_N_* = 6.76 × 10^−9^ cm^2^ s^−1^ was taken, which is the dimensionless Henry’s constant for oxygen solubility in water at 80 °C, multiplied by the oxygen diffusivity in bulk Nafion of 10^−6^ cm^2^ s^−1^ [39].

The product *K_H_D_N_* that resulted from the fitting is nearly constant as the cell current density increases (Figure 6). The only imaginable difference between the low- and high-current cell operation is the amount of liquid water in the catalyst layer, which increased with the cell operating current. The authors [40,41,42,43,44] measured the ionomer film oxygen diffusivity *D_N_* using electrochemical methods and reported *D_N_* and O_2_ permeability growth with the relative humidity, which agreed with the classic works of T. Sakai and K. Broka [45,46,47].

Moreover, O_2_ mass transport resistance was shown to decrease with inlet gas humidification, and this effect was more pronounced during transition from dry conditions to 50% RH, while a further increase in reagent inlet humidification to 100% RH resulted in a slight reduction of this parameter [7,9,19,24,48,49,50]. In our case, no significant variation of *K_H_D_N_* with the current was detected, most likely due to the chosen operating conditions when fully humidified H_2_ was supplied to the anode and 50% RH air was fed to the cathode, ensuring proper hydration of both electrodes.

A single-pore oxygen transport model gives the following formula for the transport resistivity of ionomer film [35]:(8)$$R_{N}=\frac{R_{p}l_{N}}{2K_{H}D_{N}l_{t}}$$

The film transport resistivity $R_{N}$ in the CCL was calculated using Equation (8) with the mean over the pore size distribution *R_p_*:(9)$$R_{p}=\sum_{k=1}^{9}w_{k}R_{p,k}$$
here, $R_{p,k}$ is the radius of the *k*-th pore in the spectrum. The determined film resistivity is close to the literature data of $R_{N}$ ≃ 0.3 s cm^−1^, corresponding to Pt loading of 0.1 mg_Pt_ cm^−2^ [3,4,9]. In the studied range of currents, $R_{N}$ is found to be nearly constant at ≃0.4 s cm^−1^ (Figure 7).

The effects of operating currents on oxygen transport resistance was firstly evaluated by D. Caulk and D. Baker using the limiting current method [51,52]. The authors reported that the oxygen transport resistance was constant for current density below 1 A cm^−2^, after which it increased rapidly with current density until 1.6–2.0 A cm^−2^, where it reached a second plateau. The observation was explained by the water production rate and its condensation to liquid at the cathode. The initial constant transport resistance corresponded to dry conditions when all produced water could be removed from the cathode in vapor form. As the current density increased, the water vapor pressure reached saturation, and water started to condense in the CCL porous structure leading to a growth of the transport resistance. At some point, the transport resistance reached the second plateau, indicating the maximum level of liquid saturation of the cathode electrode when the produced water could be removed mainly in liquid form. These results were confirmed by using *in-situ* neutron imaging [53] and synchrotron X-ray radiation [54,55] showing the formation of liquid water in the CCL and linking these data with results on oxygen mass transport resistance obtained by the limiting current method.

In our case, we operated at current densities below 1.0 A cm^−2^, and we could assume that the produced water was transferred from the catalyst surface to the gas channel in vapor form and did not noticeably flood the CCL [56]. Since initially the CCL was properly humidified, and there were no conditions for flooding the ionomer, film resistivity should not have been significantly affected by the operating current. We expect that a more accurate $R_{N}$ could be obtained from impedance measurements using a more detailed approximation of an experimental PSD.

## 4. Conclusions

A numerical model for local cell impedance, taking into account oxygen transport through the ionomer covering Pt/C agglomerates in the CCL and pore size distribution in the electrode, has been developed. The model has been fitted to experimental spectra acquired from10 segments in the segmented cell setup under conditions relevant to realistic PEMFC operation. Assuming an ionomer film thickness of 10 nm, from each experimental run, the fitting returns 10 values of the product of Henry’s constant for oxygen solubility in the ionomer film *K_H_* by oxygen diffusivity *D_N_* in the film (*K_H_D_N_*). Knowing *K_H_D_N_* allows us to calculate the statistical mean and confidence interval for the oxygen transport resistivity through the ionomer film in the range of the current densities from 100 to 800 mA cm^−2^. It was determined that the $R_{N}$ remains nearly constant at 0.4 s cm^−1^ for the chosen current density values.

## Figures and Tables

**Figure 1 membranes-11-00985-f001:**
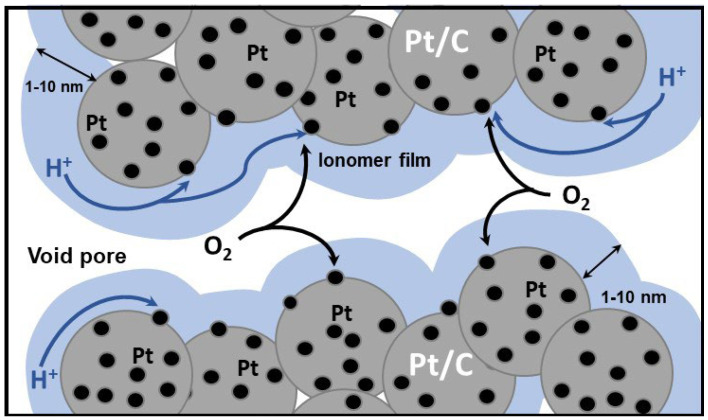
Schematic of the cathode catalyst layer structure in PEM fuel cells. Pt/C agglomerates are covered by a thin ionomer film. To reach Pt sites, oxygen transported along the void pore must penetrate through the ionomer.

**Figure 2 membranes-11-00985-f002:**
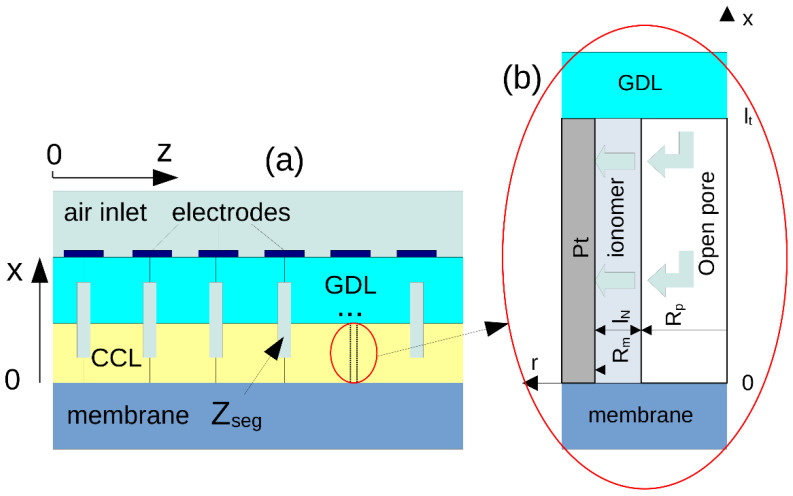
Schematic of the cell with segmented electrodes and gas diffusion layer (**a**). Sketch of a single pore used to model cathode catalyst layer impedance (**b**).

**Figure 3 membranes-11-00985-f003:**
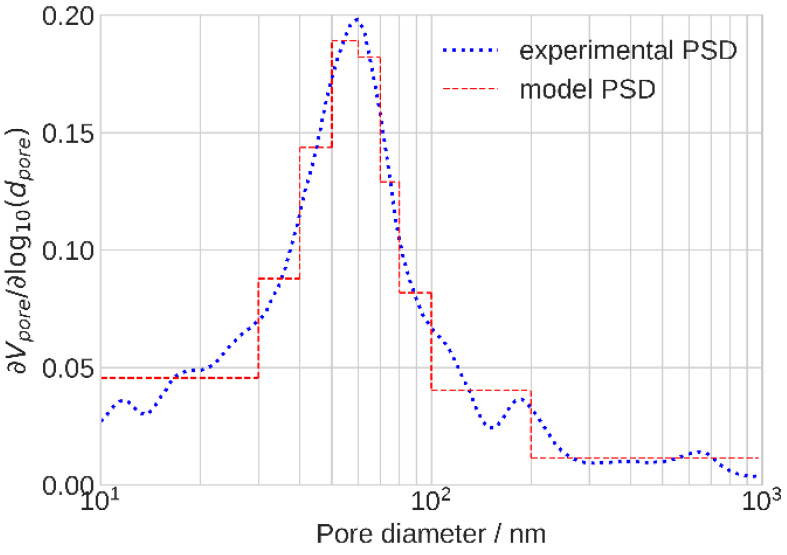
Experimental [31] (dotted line) and approximate model (dashed line) pore size distributions in the cathode catalyst layer.

**Figure 4 membranes-11-00985-f004:**
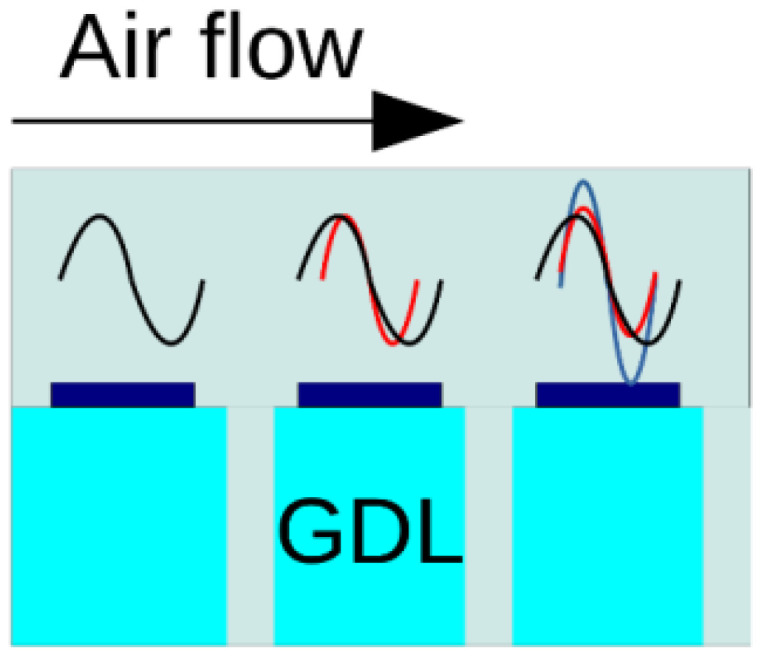
Schematic of mixing of the oxygen concentration AC perturbations $c_{h}^{1}$ due to the air flow in the channel. The perturbations $c_{h}^{1}$ produced upstream from the given segment are transported down the channel and sum up with the local perturbation. This interference affects local impedance.

**Figure 5 membranes-11-00985-f005:**
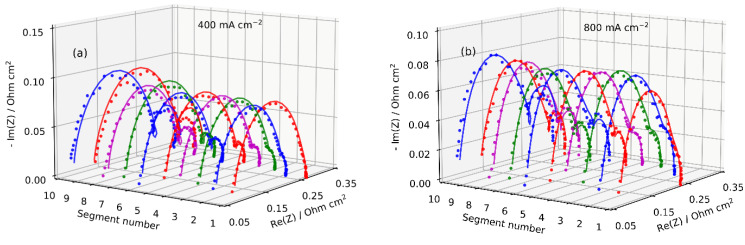
Experimental (points) and fitted model (lines) local Nyquist spectra from the individual segments at 400 (**a**) and 800 mA cm^−2^ (**b**). Air inlet is at segment 1, outlet is at segment 10.

**Figure 6 membranes-11-00985-f006:**
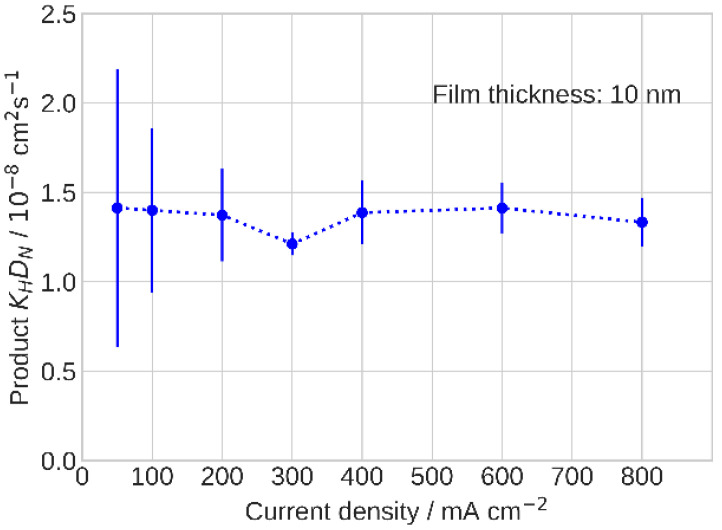
The product *K_H_D_N_* of Henry’s constant for oxygen solubility in ionomer film by oxygen diffusivity in the film vs. cell current density.

**Figure 7 membranes-11-00985-f007:**
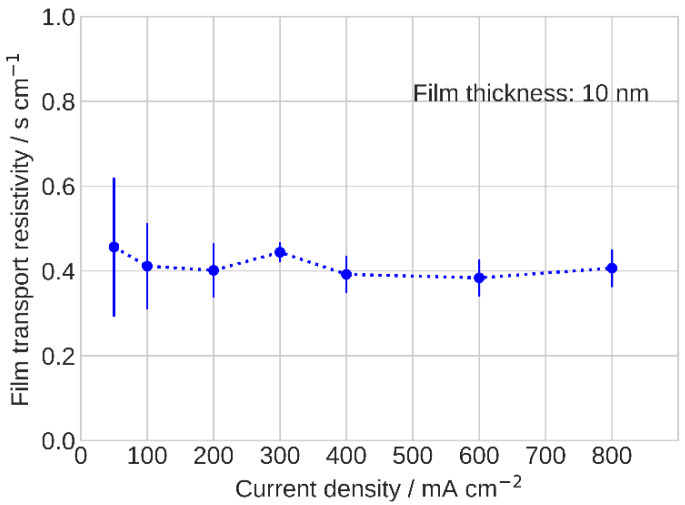
Ionomer film oxygen transport resistivity.

**Table 1 membranes-11-00985-t001:** Cell and operating parameters. A/C stands for anode/cathode.

Parameter	Value
Catalyst loading A/C, mg_Pt_ cm^−2^	0.1/0.1
CCL thickness *l_t_*, cm	3.0 × 10^−4^
GDL thickness *l_b_*, cm	230 × 10^−4^
Ionomer film thickness *l_N_*, cm	10.0 × 10^−7^
Flow stoichiometry A/C	2/4
Relative humidity A/C	100%/50%
Back pressure, kPa	150
Cell temperature, K	353

**Table 2 membranes-11-00985-t002:** Parameters of the stepwise PSD in Figure 3.

Parameter	1	2	3	4	5	6	7	8	9
Pore radius, nm	16.5	35.4	45.0	54.3	65.1	75.1	88.8	130	152
Pore volume fraction	0.199	0.100	0.127	0.137	0.112	0.0685	0.0725	0.111	0.0727

## Data Availability

The data presented in this study are available on request from the corresponding author.

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
