# Peer review of "Impedance Spectroscopy Measurements of Ionomer Film Oxygen Transport Resistivity in Operating Low-Pt PEM Fuel Cell"

_membranes, 2021, doi:10.3390/membranes11120985_

Round 1
Reviewer 1 Report
The manuscript presents a model for local impedance of a low-Pt proton exchange membrane fuel cells (PEMFCs) which includes cathode pore size distribution, O2 transport along pores and through a thin ionomer film covering Pt/C agglomerates. The model is applied to fit local impedance spectra of low-Pt fuel cells operated at current densities from 100 to 800 mA cm-2 and recorded by a 12 segmented cell system. Through fitting of electrical impedance spectroscopy data, the O2 transport resistivity through the ionomer film is obtained and compares well to the literature data.
This work is a continuation of the authors’ previous work but use a more complicated model that better simulates a real PEMFC by taking into account pore size distribution, etc. Overall, the content is clear and the results provide the reader with a quantitative understanding of the contribution in resistance from mass transport in the nanocomposite catalyst layer. It presents a considerable advance from the authors’ previous works and should be beneficial to future experiments. I do not see any obvious flaws in the arguments.
A few minor points (typos)
- In line 52, I think Khudsen should be Knudsen, if I understand it correctly.
- In line 66, “development” should be “development of” or “developing”.
Author Response
Reply to reviewers’ comments on paper membranes-1497720
“Impedance spectroscopy measurements of ionomer film oxygen transport resistivity in operating low-Pt PEM fuel cell”
Tatyana V. Reshetenko and Andrei Kulikovsky
We would like to thank the reviewer for their valuable comments. All remarks and suggestions were taken into account and the appropriate corrections are described below. Revisions made to the manuscript are marked up using the “Track Changes” function in MS Word.
Reviewer 1.
The manuscript presents a model for local impedance of a low-Pt proton exchange membrane fuel cells (PEMFCs) which includes cathode pore size distribution, O2 transport along pores and through a thin ionomer film covering Pt/C agglomerates. The model is applied to fit local impedance spectra of low-Pt fuel cells operated at current densities from 100 to 800 mA cm-2 and recorded by a 12 segmented cell system. Through fitting of electrical impedance spectroscopy data, the O2 transport resistivity through the ionomer film is obtained and compares well to the literature data.
This work is a continuation of the authors’ previous work but use a more complicated model that better simulates a real PEMFC by taking into account pore size distribution, etc. Overall, the content is clear and the results provide the reader with a quantitative understanding of the contribution in resistance from mass transport in the nanocomposite catalyst layer. It presents a considerable advance from the authors’ previous works and should be beneficial to future experiments. I do not see any obvious flaws in the arguments.
A few minor points (typos)
- In line 52, I think Khudsen should be Knudsen, if I understand it correctly.
- In line 66, “development” should be “development of” or “developing”.
We are very thankful to reviewer’s evaluation of our work.
The proper corrections were done in the text of manuscript.

Reviewer 2 Report
Fuel cells are the most promising solution to the pollution problem on the planet, as PEMFC exhaust only water and can replace other less environmentally friendly devices. This makes fuel cells a relevant topic to investigate, as fuel cell technology is imperfect and has the potential to reduce production cost. In this regard, the article has necessary scientific relevancy, as it contributes to the study of the fuel cells. The article is interesting and understandable, but there are a few comments for the authors:
1. The commercially available Gore PRIMEA membrane was used in this work, and the PSD data of these membranes were taken from the articles published in 2018. Is there more relevant PSD data for the used membranes? If so, it would be appropriate to use them for relevancy.
2. In table 1, it is unnecessary to use two units for length in lines 2, 3 and 4. Consider using only cm or only μm and nm. Also, why does the temperature is given as 273 + 80 in the last line? It would be nice to explain this in the text.
3. In item 6 Nomenclature, there are some letterings missing, for example, â„›? is not given. It would be nice to include all lettering in the table for consistency. In addition, it would be nice to supplement this item with the abbreviations presented in the article, since the article is quite full of them.
4. There are some typos in the article. For example, the character “]” is missing on line 34, it says “Khudsen” instead of “Knudsen” on line 52, it says “air ow” instead of “air flow” on line 190 and point 5 is missing
Author Response
Reply to reviewers’ comments on paper membranes-1497720
“Impedance spectroscopy measurements of ionomer film oxygen transport resistivity in operating low-Pt PEM fuel cell”
Tatyana V. Reshetenko and Andrei Kulikovsky
We would like to thank the reviewer for their valuable comments. All remarks and suggestions were taken into account and the appropriate corrections are described below. Revisions made to the manuscript are marked up using the “Track Changes” function in MS Word.
.Reviewer 2.
Fuel cells are the most promising solution to the pollution problem on the planet, as PEMFC exhaust only water and can replace other less environmentally friendly devices. This makes fuel cells a relevant topic to investigate, as fuel cell technology is imperfect and has the potential to reduce production cost. In this regard, the article has necessary scientific relevancy, as it contributes to the study of the fuel cells. The article is interesting and understandable, but there are a few comments for the authors:
- The commercially available Gore PRIMEA membrane was used in this work, and the PSD data of these membranes were taken from the articles published in 2018. Is there more relevant PSD data for the used membranes? If so, it would be appropriate to use them for relevancy.
Pore size distribution (PSD) was determined using mercury intrusion porosimetry (MIP). University of Hawaii does not have equipment to perform such evaluation of Gore PRIMEA catalyst coated membranes (CCMs), because of that we used published data for our work. Operation of the membrane electrode assemble during short period of time (below 100 hours) typically does not noticeably affect textural and morphological properties of electrodes. So, PSD results of fresh CCM and after EIS measurements obtained by MIP method should be close and relevant for the model presented in the paper.
- In table 1, it is unnecessary to use two units for length in lines 2, 3 and 4. Consider using only cm or only μm and nm. Also, why does the temperature is given as 273 + 80 in the last line? It would be nice to explain this in the text.
We revised Table 1 and changed units.
- In item 6 Nomenclature, there are some letterings missing, for example, â„›? is not given. It would be nice to include all lettering in the table for consistency. In addition, it would be nice to supplement this item with the abbreviations presented in the article, since the article is quite full of them.
The Nomenclature was revised and Abbreviation section was added.
- There are some typos in the article. For example, the character “]” is missing on line 34, it says “Khudsen” instead of “Knudsen” on line 52, it says “air ow” instead of “air flow” on line 190 and point 5 is missing.
We are very thankful to the reviewer for corrections. The proper changes were done in the manuscript.